# Osteogenic Differentiation of Human Periodontal Ligament Stromal Cells Influences Their Immunosuppressive Potential toward Allogenic CD4^+^ T Cells

**DOI:** 10.3390/ijms242216439

**Published:** 2023-11-17

**Authors:** Oliwia Miłek, Dino Tur, Lucia Ahčin, Olha Voitseshyna, Christian Behm, Oleh Andrukhov

**Affiliations:** 1Competence Center for Periodontal Research, University Clinic of Dentistry, Medical University of Vienna, 1090 Vienna, Austria; oliwia.milek@meduniwien.ac.at (O.M.); ahcinlucia@gmail.com (L.A.); a11740470@unet.univie.ac.at (O.V.); 2Clinical Division of Periodontology, University Clinic of Dentistry, Medical University of Vienna, 1090 Vienna, Austria; dino.tur@meduniwien.ac.at

**Keywords:** periodontal ligament mesenchymal stem cells, immunomodulation, osteogenic differentiation, CD4^+^ T cell proliferation, indoleamine 2,3 dioxygenase

## Abstract

The differentiation ability of human periodontal ligament mesenchymal stromal cells (hPDL-MSCs) in vivo is limited; therefore, some studies considered strategies involving their pre-differentiation in vitro. However, it is not known how the differentiation of hPDL-MSCs influences their immunomodulatory properties. This study investigated how osteogenic differentiation of hPDL-MSCs affects their ability to suppress CD4^+^ T-lymphocyte proliferation. hPDL-MSCs were cultured for 21 days in osteogenic differentiation or standard culture media. Allogeneic CD4^+^ T lymphocytes were co-cultured with undifferentiated and differentiated cells in the presence or absence of interferon (IFN)-γ, interleukin (IL)-1β or tumor necrosis factor (TNF)-α, and their proliferation and apoptosis were measured. Additionally, the effects of these cytokines on the expression of immunomodulatory or pro-inflammatory factors were investigated. Our data show that osteogenic differentiation of hPDL-MSCs reduced their ability to suppress the proliferation of CD4^+^ T lymphocytes in the presence of IFN-γ and enhanced this ability in the presence of IL-1β. These changes were accompanied by a slightly decreased proportion of apoptotic CD4^+^ in the presence of IFN-γ. The osteogenic differentiation was accompanied by decreases and increases in the activity of indoleamine-2,3-dioxygenase in the presence of IFN-γ and IL-1β, respectively. The basal production of interleukin-8 by hPDL-MSCs was substantially increased upon osteogenic differentiation. In conclusion, this study suggests that pre-differentiation strategies in vitro may impact the immunomodulatory properties of hPDL-MSCs and subsequently affect their therapeutic effectiveness in vivo. These findings provide important insights for the development of MSC-based therapies.

## 1. Introduction

The International Society of Cell and Gene Therapy defines the minimal criteria for mesenchymal stromal/stem cells (MSCs) as plastic adherence, expressing specific surface markers, and the ability to differentiate into osteoblasts, adipocytes, and chondrocytes in vitro [1,2]. These cells were initially found in the bone marrow, but nowadays, it is clear that MSCs-like cells are present in most adult tissues [3], including various dental tissues [4]. An important feature of these cells is their immunomodulatory ability [5,6]. The interaction of dental MSCs with the immune system occurs in a reciprocal way: on the one hand, they regulate immune cell activity predominantly in an immunosuppressive manner, and on the other hand, their immunomodulatory potential is strongly enhanced by inflammatory cytokines produced by immune cells [5]. Since their discovery, MSCs have been considered a promising tool in regenerative medicine [7,8]. However, despite the multilineage differentiation potential of MSCs in vitro, their differentiation in vivo is minimal and probably does not account for their therapeutic effects in vivo [9]. The beneficial effect of MSCs in vivo is achieved rather through the production of extracellular vesicles containing numerous bioactive factors and their immunomodulatory ability [9,10].

Human periodontal ligament MSCs (hPDL-MSCs) were first isolated and described by Seo et al. in 2004 [11]. The periodontal ligament, connective tissue between the tooth and alveolar bone, derives from the cranial neural crest [12]. Therefore, hPDL-MSCs, besides common trilineage differentiation potential, might also differentiate into neurons [13]. The most investigated immunomodulatory capacities of hPDL-MSCs are their abilities to suppress the proliferation of peripheral blood mononuclear cells (PBMCs) and CD4^+^ T cells, inhibit Th17 and induce Treg differentiation, and drive the polarization of macrophages toward an anti-inflammatory M2 phenotype [8,14,15,16,17]. The exact immunomodulatory abilities of hPDL-MSCs depend on the environment: priming with interferon (IFN)-γ, tumor necrosis factor (TNF)-α, or interleukin (IL)-1β results in qualitatively different immunomodulatory potential [18]. However, under certain environmental conditions, hPDL-MSCs might also exert pro-inflammatory effects [5,18]. Furthermore, they express all types of Toll-like receptors and respond to stimulation with bacterial and viral components by producing predominantly pro-inflammatory cytokines and chemokines, thereby promoting inflammatory reactions [19,20,21].

Osteogenic differentiation in vitro is one of the basic properties of MSCs [22]. This differentiation is usually induced by the prolonged culturing in specific media containing dexamethasone, ascorbic acid, and β-glycerophosphate and enhanced content of fetal bovine serum (FBS) [23,24]. Although osteogenic differentiation in vitro is a rather artificial process and does not reflect an in vivo situation entirely [25], it is a useful tool for investigating the regulation of osteogenesis by various factors. Furthermore, one potential strategy for the clinical application of MSCs for bone regeneration is their seeding on a scaffold with a topography that promotes osteogenesis or directing them to osteogenic differentiation prior to transplantation [26,27].

Since the regenerative properties of MSCs are largely mediated by their immunomodulatory function in vivo, it is important to know the relationship between osteogenic differentiation and immunomodulation in MSCs. However, this has been addressed only by a limited number of studies to date. Osteogenic differentiation of bone marrow MSCs resulted in enhancement of their ability to stimulate CD4^+^ cell migration and weakened capacity to induce the differentiation of Treg [28]. In addition, osteogenic differentiation of MSCs also resulted in increased production of IL-8 [28,29]. Finally, osteogenically differentiated bone marrow MSCs derived from ankylosing spondylitis patients, which are characterized by enhanced osteogenesis, promoted the polarization of macrophages toward the pro-inflammatory M1 state [30].

The aim of the present study was to assess how the osteogenic differentiation of hPDL-MSCs affects their immunomodulatory ability toward CD4^+^ T cells. To address this problem, osteogenic differentiation was induced in hPDL-MSCs. Afterward, they were co-cultured with allogenic CD4^+^ T cells in the presence or absence of IFN-γ, TNF-α, or IL-1β, and proliferation and apoptosis of CD4^+^ T cells were assessed. Undifferentiated hPDL-MSCs served as controls. In addition, the gene and protein expression levels of the main factors involved in immunosuppressive and inflammatory functions were evaluated.

## 2. Results

### 2.1. Osteogenic Differentiation of hPDL-MSCs

The differentiation was confirmed using Alizarin Red stain, which selectively binds to calcium ions, such as those found in mineralized tissues. Exemplary Alizarin Red stains and their quantification for all donors included in the study are presented in Figure 1. hPDL-MSCs cultured in the osteogenic differentiation media exhibited calcification (Figure 1A), which was not observed for the cells cultured in standard growth media. The OD values of cetylpyridinium chloride extracts from osteogenically differentiated cells were significantly higher than those of undifferentiated cells (Figure 1B).

### 2.2. Allogenic CD4^+^ T Cell Proliferation and Apoptosis in Co-Culture with hPDL-MSCs

Figure 2 shows the proliferation and apoptosis of allogenic CD4^+^ T cells stimulated by PHA in co-culture with undifferentiated or osteogenically differentiated hPDL-MSCs at different inflammatory milieus. The exemplary flow cytometry histograms of CFSE-stained CD4^+^ T cells (Figure 2A) showed that in the presence of IFN-γ, more generations of CD4^+^ cells were observed if they were co-cultured with osteogenically differentiated hPDL-MSCs than with undifferentiated cells. Under this condition, the percentage of CD4^+^ T cells that have divided at least once (Figure 2B) was found to be about 74% for co-culture with osteogenically differentiated hPDL-MSCs and about 57% when co-cultured with undifferentiated cells. The opposite results were found when co-culture was performed in the presence of 5 ng/mL of IL-1β: about 59% of CD4^+^ T cells divided at least once when co-cultured with osteogenically differentiated hPDL-MSCs, and about 71% when co-cultured with undifferentiated cells. No differences in CD4^+^ T cell proliferation were observed in co-cultures of control and differentiated hPDL-MSCs upon treatment with 10 ng/mL of TNF-α or in the absence of any cytokine. No statistically significant differences were found in the amount of apoptotic (dead) CD4^+^ T cells in the co-cultures with control and osteogenically differentiated hPDL-MSCs (Figure 2C). However, in the presence of IFN-γ, a tendency for a slightly higher proportion of apoptotic CD4^+^ T cells was observed upon co-culture with undifferentiated cells compared with osteogenically differentiated hPDL-MSCs (*p* = 0.14).

### 2.3. Expression Levels of Immunomodulatory Genes

The expression levels of *IDO1*, *PTGS2*, and *PD-L1* in undifferentiated and osteogenically differentiated hPDL-MSCs under basal conditions and in the presence of various inflammatory cytokines are presented in Figure 3. No statistically significant differences in the basal expression of these immunomodulatory genes were observed between undifferentiated and osteogenically differentiated cells; however, a downregulation tendency was observed for the *PD-L1* gene upon osteogenic differentiation (0.3-fold change vs. control). When stimulated with IFN-γ and TNF-α, the gene expression of *IDO1* was significantly reduced in osteogenically differentiated hPDL-MSCs compared with undifferentiated cells, showing thirteen- and twenty-times lower *IDO1* expression in differentiated cells, respectively. The expression of *PTGS-2* was not affected by osteogenic differentiation under any condition. Slightly lower *PD-L1* gene expression levels were observed in differentiated cells in the presence of IFN-γ (12 vs. 29-fold change) and TNF-α (1 vs. 4-fold change).

### 2.4. IDO1 Enzyme Activity

The activity of the IDO1 enzyme, characterized based on kynurenine production in undifferentiated and osteogenically differentiated hPDL-MSCs under different conditions, is shown in Figure 4. In the presence of IFN-γ, differentiated cells exhibited significantly lower IDO1 activity than undifferentiated cells. In the presence of IL-1β, significantly higher IDO1 activity in osteogenically differentiated hPDL-MSCs was observed. No differences in IDO1 activity were found between culture media control and differentiated hPDL-MSCs upon treatment with TNF-α or at the basal level in the absence of any cytokine.

### 2.5. Gene Expression and Protein Levels of Pro-Inflammatory Mediators

Figure 5 shows the gene expression levels and protein production of IL-6, IL-8, and MCP-1 in undifferentiated and osteogenically differentiated hPDL-MSCs with and without stimulation by inflammatory cytokines. Under the basal conditions, the gene expression levels of *IL-8* and *MCP-1* in differentiated cells were significantly upregulated, with 14- and 3-fold changes, respectively (Figure 5A). At the protein level (Figure 5B), this was confirmed only for IL-8 (53 pg/mL and 1 pg/mL in osteogenically differentiated and undifferentiated cells, respectively). In the presence of various cytokines, no significant differences in the expression of any investigated genes were found. However, at the protein level, a higher production of IL-8 by osteogenically differentiated cells than by undifferentiated cells was also observed in the presence of IFN-γ (46 pg/mL vs. 6 pg/mL, respectively). Additionally, MCP-1 production was significantly lower in osteogenically differentiated hPDL-MSCs in comparison with undifferentiated cells upon treatment with TNF-α (12,359 pg/mL vs. 26,923 pg/mL, respectively).

## 3. Discussion

In the present study, we investigated how osteogenic differentiation of hPDL-MSCs influences their immunomodulatory potential. We cultured cells either in an osteogenic induction media or a normal culture media for 23 days and confirmed the osteogenic differentiation by Alizarin Red staining. Further, we compared the basal and cytokine-induced expression of various pro-inflammatory and immunomodulatory proteins and the ability to suppress the proliferation of allogenic CD4^+^ T cells.

The main finding of the present study is that the ability of hPDL-MSCs to suppress the proliferation of allogeneic CD4^+^ T cells is affected by osteogenic differentiation depending on the inflammatory environment. Particularly, osteogenic differentiation resulted in the suppression of this ability in the presence of IFN-γ and its promotion in the presence of IL-1β. Inflammatory cytokines were used because the immunomodulatory ability of resting hPDL-MSCs is low, and it should be activated by the inflammatory environment [5]. Moreover, the mechanisms underlying the immunomodulatory abilities of hPDL-MSCs also depend on the pro-inflammatory environment [5,18].

The effect of osteogenic differentiation on the immunomodulatory activity of hPDL-MSCs is most probably associated with the alteration in IDO1 expression and activity. This enzyme catalyzes the catabolism of L-tryptophan into L-kynurenine, and the resulting depletion of tryptophan leads to the suppression of T-cell activity [31,32]. The expression of IDO1 in hPDL-MSCs is regulated by different inflammatory cytokines in different manners, with IFN-γ inducing the highest IDO1 activity [33,34]. Osteogenic differentiation resulted in the suppression of IDO1 activity in the presence of IFN-γ and its activation in the presence of IL-1β. These alterations in IDO1 activity were in line with the changes in the immunosuppressive effect of hPDL-MSCs on CD4^+^ T cell proliferation. Interestingly, a previous study showed that IDO1 deletion resulted in impaired osteogenesis of bone marrow MSCs and osteopenia in IDO1-deficient mice [35]. The direct effect of IDO1 on osteogenesis could be an important factor for bone regeneration after MSCs transplantation, besides its immunomodulatory effect.

We did not observe any essential effect of osteogenic differentiation of hPDL-MSCs on their ability to induce CD4^+^ T cell apoptosis. However, a tendency for lower CD4^+^ T cell apoptosis in the co-culture of osteogenically differentiated hPDL-MSCs in the presence of IFN-γ compared with undifferentiated cells was observed. This observation is in agreement with the lower inhibitory effect of osteogenically differentiated cells on CD4^+^ T cell proliferation in the presence of IFN-γ. This is because inhibition of proliferation could be achieved by promoting apoptosis. Slightly increased CD4^+^ T cell apoptosis under these conditions could be also related to IDO1 activity as kynurenine is known to induce CD4^+^ T cell apoptosis [36,37].

We investigated the expression of two further factors involved in the immunomodulatory function of MSCs, namely, PD-L1 and PTGS-2. PD-L1 affects CD4^+^ T cell proliferation mainly by inducing their apoptosis through the PD-1 receptor on their surface [38,39]. *PTGS-2* gene codes PGE-2 protein, which is also a well-known inhibitor of CD4^+^ T cell proliferation and differentiation [40,41]. The gene expression of *PD-L1*, another factor involved in the immunomodulatory ability of MSCs, was also affected by osteogenic differentiation. Particularly, the basal, IFN-γ- and TNF-α-induced gene expression levels of *PD-L1* were significantly decreased in the osteogenically differentiated hPDL-MSCs. However, a contribution of this factor to the decrease in proliferation of allogeneic CD4^+^ T cells observed in our study is unlikely. This is because PD-L1 is a membrane protein and can affect CD4^+^ T cell proliferation only in direct contact between these two cell types [18,42], whereas we used an experimental model without contact between these cells. We further investigated the expression of the *PTGS-2* gene, which codes PGE-2, an important factor inhibiting CD4^+^ T-cell proliferation [43]. However, we did not find any difference in *PTGS-2* expression between differentiated and undifferentiated hPDL-MSCs. Therefore, the effect of osteogenic differentiation on CD4^+^ T cell proliferation could be related mainly to the alteration in IDO1 activity.

Osteogenic differentiation was also accompanied by an increased basal production of IL-8 by hPDL-MSCs. Increased production of IL-8 by osteogenically differentiated cells was also observed in the presence of IFN-γ. However, IFN-γ is a rather weak activator of IL-8 production by hPDL-MSCs, as shown by our data (Figure 5). Therefore, the increase in IL-8 production by osteogenically differentiated cells under these conditions could also be attributed to the higher basal production. IL-1β- and TNF-α-induced IL-8 production by hPDL-MSCs was not affected by osteogenic differentiation. A recent study also observed increased IL-8 production after osteogenic differentiation of bone marrow MSCs [28]. The major function of IL-8 is the attraction of neutrophils [44,45], and therefore, osteogenically differentiated cells could create a more pro-inflammatory environment after transplantation.

The mechanisms of how osteogenic differentiation influences the immunomodulatory potential of hPDL-MSCs are unknown. A recent study showed that osteogenic differentiation of bone marrow MSCs promotes the activation of the c-Jun N terminal kinase pathway [28]. However, it is unlikely that this pathway is involved in the regulation of the suppressive effect of hPDL-MSCs on CD4^+^ T cell proliferation. This is because c-Jun-dependent signal transduction is required for the activation of IFN-γ signaling [46,47], but we observed an inhibition of IFN-γ-induced response in hPDL-MSCs upon osteogenic differentiation. Therefore, other mechanisms which are still to be elucidated must exist.

MSCs are a promising therapeutic modality, but their application in the clinic faces numerous challenges, including their limited availability and the necessity to expend them, delivery strategies, and their short lifespan after transplantation [10,48]. Although MSCs-based therapy has shown its effectiveness, at least in pre-clinical studies, the mechanisms of how these cells work after transplantation remain obscure [49]. Transplanted MSCs remain in the recipient’s tissue for only a limited time after the transplantation and are hardly engrafted, in most cases only transiently [10,50]. The immunomodulatory ability is considered one of the major mechanisms of their therapeutic effects [5,51]. Interestingly, this ability is characteristic not only for living but also apoptotic and dead cells [52]. This implies that even when MSCs die after transplantation, they can still exert their immunomodulatory effects and thus influence the regenerative processes.

Changes in the immunomodulatory ability of pre-differentiated MSCs could affect their clinical effectiveness. We cannot definitely conclude if such a strategy would enhance the clinical efficacy of transplanted cells, and based on initial observations, this strategy is not promising for the clinic. The first reason is that osteogenically differentiated hPDL-MSCs produce higher levels of IL-8 and, thus, promote inflammatory reactions. Second, depending on the environment, they also might exhibit impaired immunomodulatory ability toward CD4^+^ T cells. However, these reactions only cover part of the spectrum of the effects of transplanted cells in vivo. Further studies will be necessary to investigate the effects of osteogenic differentiation and other factors on the immunomodulatory ability toward other cells, like macrophages, and the ability of transplanted cells to survive after transplantation.

This study has some limitations. We included a cohort of diverse donors, but the influence of donor-to-donor variability calls for cautious interpretation as the findings may only be partially universal to the broader human population. This concern could be addressed by the inclusion of an expanded donor pool, enhancing the robustness and applicability of the outcomes. We used the same cell number for both conditions at the beginning of the experiment but did not control the total cell number at the end of 23 days of culture. However, under the microscope, the cells were fully confluent, and therefore, the differences between the two culture conditions are not expected to be critical. Moreover, using indirect co-culture conditions introduces limitations in researching reciprocal interactions between hPDL-MSCs and CD4^+^ T cells. Direct co-culture methodologies could be considered for future investigations. Furthermore, additional time points preceding the final endpoint could provide insight during earlier stages of osteogenic differentiation.

## 4. Materials and Methods

### 4.1. Cell Isolation and Culture

Cells were isolated from periodontal ligaments of the molars of 11 healthy volunteers of both sexes, aged between 17 and 33, extracted for orthodontic purposes [20,33]. The usage of cells was approved by the Ethics Committee affiliated with the Medical University of Vienna (No. 1079/2019). All the procedures followed the “Good Scientific Practice” regulations of the Medical University of Vienna and the ethical principles outlined in the Declaration of Helsinki, and all participants gave their written consent. Briefly, the periodontal ligament was carefully scraped off the tooth and plated on a 10 cm tissue culture plate (Sarstedt, Nümbrecht, Germany). The explant was supplemented with Dulbecco’s modified Eagle’s medium (DMEM; Capricorn Scientific, Ebsdorfergrund, Germany), containing 10% fetal bovine serum (FBS, Capricorn Scientific, Ebsdorfergrund, Germany), 100 I.U./mL penicillin and 100 µg/mL streptomycin (Pen/Strep, Capricorn Scientific, Ebsdorfergrund, Germany). The cells were cultured at 37 °C with 5% CO_2_ under humidified conditions. Once human periodontal ligament MSCs (hPDL-MSCs) outgrew from the explant and reached 80% confluence, they were passaged and maintained in cell culture flasks (Sarstedt, Nümbrecht, Germany) under the same culture conditions as mentioned above. Cells up to passage seven were used in the study.

### 4.2. Osteogenic Differentiation and Alizarin Red Staining

hPDL-MSCs were seeded onto 12- and 24-well cell culture plates (TPP, Trasadingen, Switzerland) at a density of 2 × 10^4^ cells per cm^2^ in 1 mL or 500 µL, respectively, of culture medium (DMEM + 10% FBS + 100 I.U./mL penicillin and 100 µg/mL streptomycin) and cultured for one week. On day 7, osteogenic differentiation was initiated: the cells were washed with 1× PBS (Gibco, Waltham, MA, USA) and then supplemented with 500 µL of the osteogenic medium consisting of αMEM (Capricorn Scientific, Ebsdorfergrund, Germany), and containing 10% FBS, 100 I.U./mL penicillin, 100 µg/mL streptomycin, 0.1 μM dexamethasone (Sigma-Aldrich, St. Louis, MO, USA), 50 μM ascorbic acid 2-phosphate (Sigma-Aldrich, St. Louis, MO, USA), and 10 mM β-glycerophosphate (Sigma-Aldrich, St. Louis, MO, USA). The control cells were further cultured in a culture medium. The cells were further cultured for 23 days; half of the medium was changed every 3–4 days. The cells cultured in both mediums were confluent after 23 days of culture, and no differences between them were observed under the light microscope.

On day 23, post the start of osteogenic differentiation, the cells were stained with Alizarin Red stain [53]. A commercially available kit was used (ARS, Sigma-Aldrich, St. Louis, MO, USA). Initially, the cells were fixed in 70% ice-cold ethanol for 45 min, followed by washing with distilled H_2_O (dH_2_O). A 1% *w/v* Alizarin Red solution was prepared in dH_2_O, filtered through a 0.45 μm filter, and used to submerge the cells (500 μL for a 24-well plate, 1 mL for a 12-well plate) for staining at room temperature. After 45 min, the dye was removed, and the cells were washed with dH_2_O. Microscopic pictures were taken to visualize calcium deposits. Subsequently, the ARS was extracted using a 10% *w/v* cetylpyridinium chloride solution (CPC, Sigma-Aldrich, St. Louis, MO, USA) in dH_2_O. The cells were submerged in the same volumes of 10% CPC as mentioned above for ARS staining and incubated for 45 min at room temperature with gentle shaking. Then, the solution was transferred to a 96-well plate, and absorbance was measured at a wavelength of 405 nm using the Synergy HTX Multimode Reader (BioTek, Winooski, VT, USA). The confirmation of osteogenesis was performed for every donor and every type of experiment. For each donor and experiment, two wells were used for the osteogenic differentiation, and the measurements of the absorbance were performed in technical triplicates.

### 4.3. Isolation of Allogenic CD4^+^ T Cell and Their Co-Culture with hPDL-MSCs

Allogeneic CD4^+^ T cells were isolated from two healthy volunteers (male 48 y.o. and male 28 y.o.). Whole blood was collected into heparin-coated vacuum vials (Greiner Bio-One, Frickenhausen, Germany) and processed further under sterile conditions. The blood was diluted 1:1 with pre-warmed HBSS (Gibco, Waltham, MA, USA) at 37 °C. Subsequently, diluted blood was carefully layered on top of Ficoll-Paque Plus (Cytiva, Marlborough, MA, USA) and centrifuged for 30 min. The layer containing PBMCs was collected and washed with HBSS. After counting, the cells were processed using the MagniSort Human CD4 T cell Enrichment Kit (Invitrogen, Waltham, MA, USA) following the manufacturer’s protocol. The resulting pure CD4^+^ T cells were stained with the CellTrace CFSE Cell Proliferation Kit (Invitrogen, Waltham, MA, USA). Briefly, 1 × 10^6^ cells/mL were resuspended in pre-warmed 1× PBS supplemented with 5% FBS. CFSE staining solution, resuspended in dimethylsulfoxide to a 5 mM concentration, was added to the cells to achieve a working concentration of 2.5 μM, followed by a 10 min incubation at room temperature. The cells were then washed and resuspended in fresh RPMI 1640 (Capricorn Scientific, Ebsdorfergrund, Germany) supplemented with 10% FBS, 100 I.U./mL penicillin, and 100 µg/mL streptomycin and incubated for an additional 10 min at 37 °C.

Co-culture of CFSE-stained allogeneic CD4^+^ T cells with hPDL-MSCs was performed using methods similar to those previously described [18]. hPDL-MSCs were cultured in 12-well plates under control and osteogenic conditions as described above. On day 23, before setting up a co-culture model, hPDL-MSCs were primed for 24 h with human recombinant inflammatory cytokines (PeproTech, London, UK; IFN-γ (100 ng/mL), TNF-α (10 ng/mL), or IL-1β (5 ng/mL)) in 1 mL DMEM supplemented with 100 I.U./mL penicillin and 100 µg/mL streptomycin. The following day, the medium for hPDL-MSCs was changed to RPMI 1640 supplemented with 10%FBS, 100 I.U./mL penicillin and 100 µg/mL streptomycin and containing IFN-γ, TNF-α, or IL-1β at the concentrations mentioned above. Then, 12-well 0.4 μm pore inserts (Sarstedt, Nümbrecht, Germany) containing 4 × 10^5^ of CFSE-stained CD4^+^ T cells in 500 µL of complete RPMI 1640 were placed on top of the hPDL-MSC layer. Then, 10 μg/mL phytohemagglutinin (PHA, Gibco, Waltham, MA, USA) was added to the T cells to induce their proliferation.

### 4.4. Analysis of CD4^+^ T Cell Proliferation and Apoptosis by Flow Cytometry

After 5 days of co-culture, CD4^+^ T cells were collected into FACS tubes, the media were removed by centrifugation, and the cells were resuspended in 100 µL of FACS buffer (which consisted of PBS supplemented with 3% BSA and 0.09% NaN3) and 10 μL of propidium iodide (PI, Invitrogen, Waltham, MA, USA), and incubated for 5 min. Immediately afterward, CD4^+^ T cell proliferation and apoptosis were evaluated using the Attune NxT Flow Cytometer (Invitrogen, Waltham, MA, USA) based on the appearance of new generations of CFSE-labelled cells and PI-positive cells, respectively. After excluding doublets, the number of dead (apoptotic) cells and the number of live cells that had undergone at least one round of proliferation were measured. The results were analyzed using the Attune NxT Flow Cytometer Software v3.1.2 and FCS Express Flow Cytometry Analysis Software (De Novo Software, Pasadena, CA, USA).

### 4.5. Indoleamine-2,3-Dioxygenase (IDO1) Activity Assay

hPDL-MSCs were either osteogenically differentiated or cultured in a culture medium in 12-well plates using the methods described above. On day 23 after the start of osteogenic differentiation, the cells were stimulated for 72 h with human recombinant inflammatory cytokines (PeproTech, London, UK): IFN-γ (100 ng/mL), TNF-α (10 ng/mL) or IL-1β (1 ng/mL). This was performed in 1 mL of DMEM supplemented with 100 I.U./mL penicillin, and 100 µg/mL streptomycin. Then, the IDO1 activity assay was performed: 400 µL of the supernatant was harvested, mixed with 200 µL of 30% trichloroacetic acid (*w/v* in dH_2_O) and incubated at 65 °C for 30 min. After incubation, centrifugation at 10,000 rpm for 5 min separated the precipitated proteins from the supernatant. For the subsequent analysis, 125 µL of the supernatant, combined with an equal volume of Ehrlich’s reagent (200 mg of P-dimethylbenzaldehyde with 10 mL of acetic acid), was transferred to a 96-well plate, with duplicates of each sample, followed by shaking and incubation at room temperature in a dark environment for 10 min. The absorbance of the samples was then measured at 492 nm in a Synergy HTX Multimode Reader (BioTek, Winooski, VT, USA). A standard curve of defined L-kynurenine concentrations in DMEM supplemented with 100 I.U./mL penicillin and 100 µg/mL streptomycin was established. The L-kynurenine concentration was determined by comparing the absorbance values of samples with the known concentrations established in the standard curve.

### 4.6. Reverse Transcription-Quantitative Polymerase Chain Reaction (RT-qPCR)

hPDL-MSCs were cultured in 24-well plates using 4 × 10^4^ of initially seeded cells per well and 500 µL of corresponding media. On day 21 post the start of differentiation, cells were stimulated for 24 h with IFN-γ (100 ng/mL), TNF-α (10 ng/mL) or IL-1β (1 ng/mL) (PeproTech, London, UK) in DMEM supplemented with 100 I.U./mL penicillin, and 100 µg/mL streptomycin. After stimulation, the cells were lysed and mRNA was reverse transcribed to cDNA using a TaqMan Gene Expression Cells-to-Ct Kit (Invitrogen, Waltham, MA, USA), according to manufacturer’s protocol, and supernatants were collected for the protein analysis. cDNA generation was performed using a Biometra TOne PCR Thermal Cycler (Analytik Jena, Jena, Germany) at the following settings: 37 °C for 1 h, 95 °C for 5 min, followed by an indefinite 4 °C. The quantitative polymerase chain reaction was performed using TaqMan Gene Expression Assays (Applied Biosystems, Waltham, MA, USA) with the following ID numbers: Hs00985639_m1 IL-6, Hs00174103_m1 IL-8, Hs00234140_m1 monocyte chemoattractant protein 1 (MCP-1)/CCL2, Hs00984146_m1 IDO1, Hs00153133_m1 prostaglandin-endoperoxide synthase 2 (PTGS2), Hs00204257_m1 programmed death-ligand 1 (PD-L1)/CD274 and Hs0099999905_m1 GAPDH. The reaction was set in a StepOnePlus Real-Time PCR System (Applied Biosystems, Waltham, MA, USA) with the following cycling conditions: 95 °C for 10 min, followed by 50 cycles of denaturation at 95 °C for 15 s and annealing and elongation at 60 °C for 1 min. Gene expression was quantified by the 2^−ΔΔCt^ method using the formula ΔΔC_t_ = (*C*_t_^target^ − *C*_t_^GAPDH^)_sample_ − (*C*_t_^target^ − *C*_t_^GAPDH^)_control_, taking GAPDH as a reference gene and cells cultured in culture media without inflammatory cytokines as the control.

### 4.7. Enzyme-Linked Immunosorbent Assay (ELISA)

The supernatants collected prior to the cell lysis for RT-qPCR were centrifuged to remove cellular debris. Protein level quantification was performed with an IL-6 Human Uncoated ELISA Kit, an IL-8 Human Uncoated ELISA Kit, and an MCP-1/CCL2 Human Uncoated ELISA Kit (Invitrogen, Waltham, MA, USA), according to the manufacturer’s protocols. The optical density (OD) was measured at 450 nm and 570 nm in a Synergy HTX Multimode Reader (BioTek, Winooski, VT, USA). OD_570_ was subtracted from OD_450_, and sample concentrations were calculated based on the standard curve using arigo’s ELISA Calculator (arigo Biolaboratories, Hsinchu, Taiwan). All values below the detection level were considered zero.

### 4.8. Statistical Analysis

hPDL-MSCs isolated from 11 different donors were used in this study. Each experiment was performed with the cells of at least three different donors with two to three technical replicates. GraphPad Prism 8 (Dotmatics, Boston, MA, USA) was used to analyze the data and produce the graphs. Normal distribution was evaluated with the Shapiro–Wilk test. The differences between groups were analyzed with a paired *t*-test for normally distributed data or a Wilcoxon signed-rank test in other cases. *p* values  ≤  0.05 were considered statistically significant.

## 5. Conclusions

The data of the present study show that osteogenic differentiation of hPDL-MSCs results in alterations in their immunomodulatory properties, particularly the ability to suppress CD4^+^ T cell proliferation. Like the immunomodulatory properties of MSCs in general, the effect of osteogenic differentiation depends on the inflammatory environment and could be associated with altered IDO1 activity. In addition, increased production of IL-8 by osteogenically differentiated cells might contribute to creating a pro-inflammatory environment. A potential alteration in the immunomodulatory properties should always be considered when developing new strategies for MSC-based therapy.

## Figures and Tables

**Figure 1 ijms-24-16439-f001:**
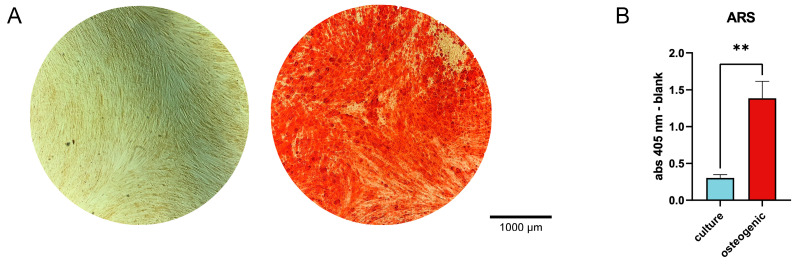
Osteogenic differentiation of hPDL-MSCs visualized with Alizarin Red staining. (**A**)—Representative microscopic pictures; photos taken with an Echo Revolve microscope, with objective 4×. (**B**)—Quantification of Alizarin Red staining. The stains were extracted using a 10% *w/v* cetylpyridinium chloride solution, and the optical density was measured at 405 nm (*Y*-axis). Data are presented as mean ± s.e.m. (*n* = 11). ** *p* ≤ 0.01.

**Figure 2 ijms-24-16439-f002:**
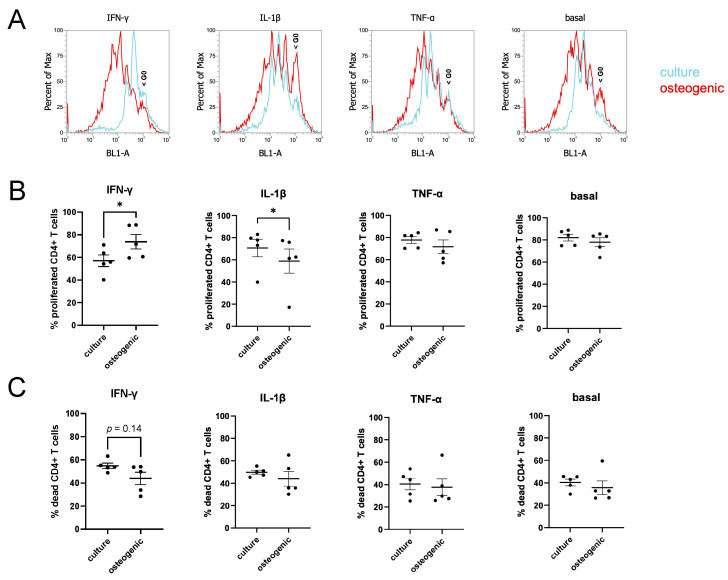
Proliferation and apoptosis of allogeneic CD4^+^ T cells in the presence of undifferentiated and osteogenically differentiated hPDL-MSCs. CD4^+^ T cells were stained with CFSE, stimulated with 10 µg/mL PHA, and co-cultured with undifferentiated or osteogenically differentiated hPDL-MSCs in the presence of IFN-γ (100 ng/mL), IL-1β (5 ng/mL), or TNF-α (10 ng/mL) or in the absence of any cytokine (basal). (**A**)—Representative overlays from one donor under different inflammatory milieus, G0—non-divided CD4 T cells. (**B**)—Quantification of CD4^+^ T cells proliferation. Y-axes show the percentage of CD4^+^ T cells that have divided at least once, measured based on CFSE staining. CFSE intensity of T cells was measured with flow cytometry. Data are presented as mean ± s.e.m. (**C**)—Dead CD4^+^ T cells at the end of co-culture quantified based on propidium iodide (PI) staining. Y-axes represent the percentage of PI-positive T cells. Data are presented as mean ± s.e.m. * *p* ≤ 0.05.

**Figure 3 ijms-24-16439-f003:**
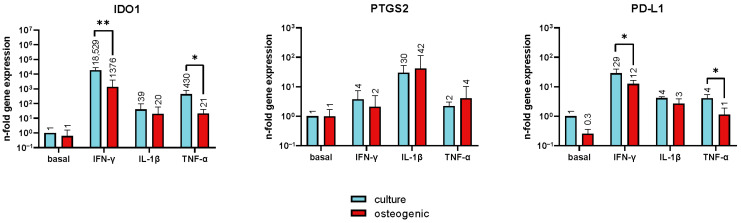
Gene expression levels of immunomodulatory factors in undifferentiated and osteogenically differentiated hPDL-MSCs induced by various inflammatory cytokines. hPDL-MSCs were grown in culture or osteogenic differentiation conditions for 22 days and stimulated with 100 ng/mL of IFN-γ, 5 ng/mL of IL-1β, or 10 ng/mL of TNF-α for 24 h, and the expression *IDO1*, *PTGS2*, and *PD-L1* genes was evaluated by RT-qPCR. Y-axes show the n-fold changes in gene expression in relation to unstimulated and undifferentiated cells. Data are presented as mean ± s.e.m. (*n* = 6). * *p* ≤ 0.05, ** *p* ≤ 0.01.

**Figure 4 ijms-24-16439-f004:**
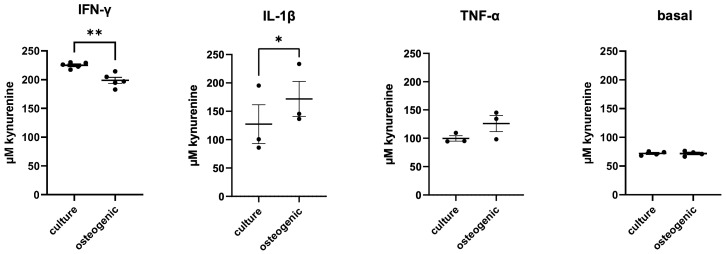
IDO1 enzyme activity of undifferentiated and osteogenically differentiated hPDL-MSCs after stimulation with various cytokines. hPDL-MSCs were grown in culture or osteogenic differentiation conditions for 22 days and then stimulated with 100 ng/mL of IFN-γ, 5 ng/mL of IL-1β, or 10 ng/mL of TNF-α, or treated without any cytokines (basal). The concentration of kynurenine, which is directly proportional to the IDO1 enzyme activity, was measured in the conditioned media. Y-axes show concentration of kynurenine, presented as mean ± s.e.m. * *p* ≤ 0.05, ** *p* ≤ 0.01.

**Figure 5 ijms-24-16439-f005:**
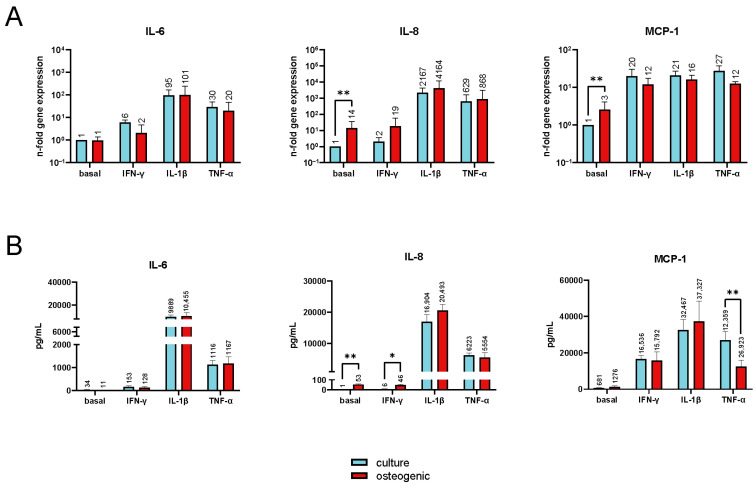
Gene expression levels and protein production of IL-6, IL-8, and MCP-1 in undifferentiated and osteogenically differentiated hPDL-MSCs. hPDL-MSCs were cultured in a standard or osteogenic differentiation media for 22 days and then stimulated with 100 ng/mL of IFN-γ, 5 ng/mL of IL-1β, or 10 ng/mL of TNF-α for 24 h. The gene expression of *IL-6*, *IL-8*, and *MCP-1* (**A**) and the production of corresponding proteins (**B**) were assessed by RT-qPCR and ELISA, respectively. Y-axes (**A**) show the n-fold change in gene expression compared with non-stimulated cells grown in culture media. The protein levels (**B**) were measured in media supernatants. Data are presented as mean ± s.e.m. (*n* = 6). * *p* ≤ 0.05, ** *p* ≤ 0.01.

## Data Availability

The data presented in this study are available on request from the corresponding author.

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
