# Peer review of "Osteogenic Differentiation of Human Periodontal Ligament Stromal Cells Influences Their Immunosuppressive Potential toward Allogenic CD4+ T Cells"

_ijms, 2023, doi:10.3390/ijms242216439_

Round 1

Reviewer 1 Report

Comments and Suggestions for Authors

In their research, O. Milek et al. explored the impact of osteogenic differentiation on the ability of human periodontal ligament mesenchymal stromal cells (hPDL-MSCs) to suppress the immune response of allogeneic CD4+ T cells. Given the limited differentiation potential of hPDL-MSCs in vivo, the study emphasized the significance of pre-differentiation techniques in vitro, though the effects on their immune-regulating properties were not well understood previously.

The methodology of the study was robust: hPDL-MSCs underwent osteogenic differentiation and were maintained in standard media for 21 days, then co-cultured with allogeneic CD4+ T lymphocytes under different cytokine environments (IFN-γ,IL-1β,TNF-α).

The advanced techniques used in this study included analysis of cell proliferation and apoptosis, along with evaluating the influence of cytokines on the expression of immunomodulatory or pro-inflammatory factors in hPDL-MSC.

The findings showed that osteogenic differentiation affected the hPDL-MSCs' capacity to inhibit the proliferation of CD4+ T lymphocytes differently in the presence of IFN-γ compared to IL-1β. Furthermore, changes in the activity of the enzyme indoleamine-2,3-dioxygenase were observed following osteogenic differentiation.

These new results hold potential clinical implications. They suggest that pre-differentiating hPDL-MSCs in vitro might alter their therapeutic effectiveness in vivo. Additionally, this study provides valuable insights for the ongoing development of MSC-based therapies.

Comments on the Quality of English Language

The quality of English is good.

Author Response

Regarding Reviewer 1

We appreciate this Reviewer for throughout positive assessment of or manuscript. The only point raised by this Reviewer was some minor English editing of our manuscript, which was done by us for the revised version.

Reviewer 2 Report

Comments and Suggestions for Authors

The authors look at the biological response of cells to media collected from differentiated and undifferentiated hPDL-MSCs. The show cells respond different to the different secreted media.  However I am not sure how this relates to the use of the cells as a therapeutic agent as why and when would they be differentiated first before transplantation and their use? Some clarity here would have been helpful, a mention was made in the introduction but I think more discussion should be made  at the end in the discussion.

.Specific comments:

1) How was standardisation made between the two conditioned media? cell number or protein concentration? This is important when differences in an effect upon treatment is being measured.

2) Surely once transplanted in vivo the properties of the cells could change again and not reflect in vitro findings?

3) Some explanation of the function of  of IDO1, PTGS2, and PD-L1 would have been helpful.

4) Could the authors validate findings ? Add an inhibitor of IDo-1, or various neutralising antibodies to proposed candidates?

5) A little more justification of what these findings mean in vivo and in specific transplantation strategies would also be helpful to clarify the reasoning behind the work.

Author Response

Regarding Reviewer 2

We are thankful to this Reviewer for the critical assessment of our manuscript and highly relevant comments

Reviewer’s comment 1

How was standardisation made between the two conditioned media? cell number or protein concentration? This is important when differences in an effect upon treatment is being measured.

Authors answer

Thank you for this important comment. In our study we tried to standardize the experimental condition as good as possible. The Reviewer correctly mentioned that some differences are expected because the different composition of media. We used the same cell number at the beginning of the culture. However, at the end of the culture for 23 days, the cells were confluent, and, therefore, we do not expect any essential differences in the total cell number. Moreover, some parameters, like gene expression levels, are independent on the total cell number. We mentioned this issue in the Material and Methods section (see, p. 10) and also in the limitation section of our manuscript (see, p. 9).

Reviewer’s comment 2

Surely once transplanted in vivo the properties of the cells could change again and not reflect in vitro findings?

Authors answer

Thank you for this comment. It is correct that it is difficult, if possible at all, to reflect in vivo situations using in vitro approaches. This aspect is mentioned in several places in our manuscript and is discussed in a new paragraph about the clinical significance of our data (see, p. 9).

Reviewer’s comment 3

Some explanation of the function of  of IDO1, PTGS2, and PD-L1 would have been helpful.

Authors answer

We added the explanation of IDO-1, PTGS-2 and PD-L1 function in the revised version of our manuscript. The explanation of IDO-1 function was provided in the paragraph discussing IDO-1 activity (see, p. 8) and the information about PTGS-2 and PD-L1

Reviewer’s comment 4

Could the authors validate findings ? Add an inhibitor of IDo-1, or various neutralising antibodies to proposed candidates?

Authors answer

Thank you for this important point. In the present study we did not plan to validate this specific issue, because our study was mainly focused on the alteration of hPDL-MSCs ability to suppress T-cell proliferation upon their osteogenic differentiation. However, this step will be considered in our future experiments. Now, we are planning a new study considering different models and different inhibitors

Reviewer’s comment 5

A little more justification of what these findings mean in vivo and in specific transplantation strategies would also be helpful to clarify the reasoning behind the work.

Authors answer

Thank you for this comment. In the revised version, we added the paragraph discussing this problem (see, p. 9).